# Effect of TiO_2_ Film Thickness on the Stability of Au_9_ Clusters with a CrO_x_ Layer

**DOI:** 10.3390/nano12183218

**Published:** 2022-09-16

**Authors:** Abdulrahman S. Alotabi, Yanting Yin, Ahmad Redaa, Siriluck Tesana, Gregory F. Metha, Gunther G. Andersson

**Affiliations:** 1Flinders Institute for Nanoscale Science and Technology, Flinders University, Adelaide, SA 5042, Australia; 2Department of Physics, Faculty of Science and Arts in Baljurashi, Albaha University, Baljurashi 65655, Saudi Arabia; 3Flinders Microscopy and Microanalysis, College of Science and Engineering, Flinders University, Adelaide, SA 5042, Australia; 4Department of Earth Sciences, University of Adelaide, Adelaide, SA 5005, Australia; 5Faculty of Earth Sciences, King Abdulaziz University, Jeddah 21589, Saudi Arabia; 6The MacDiarmid Institute for Advanced Materials and Nanotechnology, School of Physical and Chemical Sciences, University of Canterbury, Christchurch 8041, New Zealand; 7National Isotope Centre, GNS Science, Lower Hutt 5010, New Zealand; 8Department of Chemistry, University of Adelaide, Adelaide, SA 5005, Australia

**Keywords:** RF magnetron sputtering, TiO_2_ film, morphology, triphenylphosphine, Au_9_, gold clusters, photodeposition, CrO_x_, Cr(OH)_3_, Cr_2_O_3_ layer

## Abstract

Radio frequency (RF) magnetron sputtering allows the fabrication of TiO_2_ films with high purity, reliable control of film thickness, and uniform morphology. In the present study, the change in surface roughness upon heating two different thicknesses of RF sputter-deposited TiO_2_ films was investigated. As a measure of the process of the change in surface morphology, chemically -synthesised phosphine-protected Au_9_ clusters covered by a photodeposited CrO_x_ layer were used as a probe. Subsequent to the deposition of the Au_9_ clusters and the CrO_x_ layer, samples were heated to 200 ℃ to remove the triphenylphosphine ligands from the Au_9_ cluster. After heating, the thick TiO_2_ film was found to be mobile, in contrast to the thin TiO_2_ film. The influence of the mobility of the TiO_2_ films on the Au_9_ clusters was investigated with X-ray photoelectron spectroscopy. It was found that the high mobility of the thick TiO_2_ film after heating leads to a significant agglomeration of the Au_9_ clusters, even when protected by the CrO_x_ layer. The thin TiO_2_ film has a much lower mobility when being heated, resulting in only minor agglomeration of the Au_9_ clusters covered with the CrO_x_ layer.

## 1. Introduction

Titanium dioxide (TiO_2_) is a semiconductor widely used for a large range of photocatalytic applications and is also an ideal model system for various types of studies [1,2]. There are various techniques to prepare TiO_2_ films, such as sol-gel [3], evaporation [4], chemical vapour deposition [5], atomic layer deposition [6] and radio frequency (RF) magnetron sputtering [7]. Each of these methods has advantages and disadvantages in regard to fabrication costs, uniformity of the film morphology, thermal stability, purity and preparation time. Therefore, the best method of choice for TiO_2_ film preparation depends on which application the film will be used in.

Amongst the above-named methods, RF magnetron sputtering is known to produce high-purity TiO_2_ films with uniform thickness, ease of use and strong film adhesion to the substrate [8]. The properties of these films are significantly impacted by the sputtering conditions, such as RF power, gas pressure, substrate type, substrate temperature and target-to-substrate distance [9,10,11,12,13,14]. For instance, it has been reported that control of TiO_2_ film thickness is possible by modulating the deposition time and the gas sputtering pressure [15].

TiO_2_ films prepared with the RF magnetron sputtering method can be amorphous or have a rutile, anatase, or brookite crystal structure. It is well known that the physical properties of TiO_2_ films depend highly on the post-deposition treatment, including heat treatment conditions [16,17,18]. Çörekçi et al. reported that a correlation between heating treatment and surface morphology with different TiO_2_ film thicknesses. It was observed that an increase in surface roughness and grain sizes occurred during heating depending on TiO_2_ film thicknesses, which also increased with film thickness. This is because increasing temperatures transform TiO_2_ from amorphous to anatase and then to rutile [17], and these phase transitions affect the surface morphology of the TiO_2_ film, which includes the roughness and crystallinity of the surface [19].

The aim of this study is to investigate the influence of heat treatment on the surface morphology of RF sputter-deposited TiO_2_ films with two different thicknesses, and the effect this has on size-specific Au clusters deposited on the surface. TiO_2_ films have attracted interest as substrates for investigating the role of Au clusters as cocatalysts in photocatalysis [20,21]. In these studies, TiO_2_ films had been heated as part of the sample preparation procedure. The change in morphology, including surface mobility, upon heating, can lead to agglomeration of the Au clusters. Understanding the change in surface morphology upon heating, thus, is important when using TiO_2_ as a substrate for investigating the cocatalyst properties. In the present work, phosphine-protected Au_9_ clusters covered by a photodeposited CrO_x_ layer were used as probes for the TiO_2_ mobility during the change of morphology upon heating. Scanning electron microscopy (SEM), X-ray diffraction (XRD), laser scanning confocal microscope (LSCM) and X-ray photoelectron spectroscopy (XPS), have been applied to characterise the thickness, crystal structure, surface morphology and chemical composition and size of the Au cluster. The importance of the present work is to show that morphology changes in RF sputter-deposited TiO_2_ depend on the thickness of the TiO_2_ layer, and that Au_9_ clusters can be used to probe morphology changes in the surface.

## 2. Experimental Methods and Techniques

### 2.1. Material and Sample Preparation

#### 2.1.1. Preparation of TiO_2_ Films

The RF magnetron sputtering method was used to prepare TiO_2_ films on a silicon wafer under high vacuum conditions (HHV/Edwards TF500 sputter coater) [22]. Before the deposition, the silicon wafer was cleaned with ethanol and acetone and then dried in a stream of dry nitrogen. The TiO_2_ film was deposited onto a p-type silicon wafer substrate by sputtering a 99.9% pure TiO_2_ ceramic target with 500 W sputtering power using Ar^+^ (flow rate of 5 sccm) for 50 min. The sputter coating chamber was held under vacuum at 2 × 10^−5^ mbar. This process resulted in TiO_2_ films formed on the silicon wafer with a native oxide layer of TiO_2_.

TiO_2_ films with two different thickness were fabricated applying the above-described procedure. The TiO_2_ films had different colours based on light interference [23]: a TiO_2_/Si wafer with a purple colour and a TiO_2_/Si wafer with a gold-like colour (see Appendix A). The difference in colour of the films is related to the difference in light interference patterns within the films due to their difference in film thickness [24]. The thickness of TiO_2_P is ~400 nm, while TiO_2_G is ~1100 nm (*vide infra*). The TiO_2_ wafers were cut into 1 cm × 1 cm pieces and used without further treatment. The two TiO_2_ wafers are hereafter referred to as (i) TiO_2_P and (ii) TiO_2_G.

#### 2.1.2. Deposition of Au_9_ Clusters

The deposition procedure of Au_9_(PPh_3_)_8_(NO_3_)_3_ (Au_9_) was identical for both the TiO_2_P and TiO_2_G samples. Phosphine-protected Au_9_ clusters were synthesised as reported previously [25]. A UV-Vis spectrum of the Au_9_ cluster is shown in Appendix A. The TiO_2_ films were immersed in Au_9_ methanol solutions (2 mL) for 30 min at concentrations of 0.006, 0.06 and 0.6 mM. The TiO_2_ samples were rinsed by quickly dipping them into pure methanol and then dried in a stream of dry nitrogen. These samples are hereafter referred to as (i) TiO_2_P-Au_9_ and (ii) TiO_2_G-Au_9_.

#### 2.1.3. Photodeposition of CrO_x_ Layer

Photodeposition of the CrO_x_ layer was the same for both TiO_2_-Au_9_ samples (TiO_2_P-Au_9_ and TiO_2_G-Au_9_). A 0.5 mM potassium chromate solution was prepared by dissolving K_2_CrO_4_ (≥99%, Sigma-Aldrich) in deionised water. The TiO_2_-Au_9_ samples were immersed into the K_2_CrO_4_ solution (1 mL) and irradiated for 1 h using a UV LED (Vishay, VLMU3510-365-130) with ~1 cm between the sample and the irradiation source. The UV LED had a radiant power of 690 mW at 365 nm wavelength. After photodeposition, the samples were washed by dipping them into deionised water and dried in a stream of dry nitrogen [26]. These samples are hereafter referred to as (i) TiO_2_P-Au_9_-CrO_x_ and (ii) TiO_2_G-Au_9_-CrO_x_.

#### 2.1.4. Heat Treatment

To remove the phosphine ligands from Au_9_ clusters, all samples were treated with heating at 200 ℃ for 10 min under ultra-high vacuum (1 × 10^−8^ mbar) in the XPS chamber.

### 2.2. Characterization Methods

#### 2.2.1. Scanning Electron Microscopy and Energy Dispersive X-ray Spectroscopy (SEM-EDAX)

The thickness of TiO_2_ films (TiO_2_P and TiO_2_G) was determined by combining SEM imaging and SEM-EDAX (FEI Inspect F50 microscope) scans on cross-sections of the TiO_2_ samples. Cross-sectional images were recorded at a magnification of up to 100 k with 15 keV electron energy.

#### 2.2.2. X-ray Diffraction (XRD)

The crystal and phase structure of the TiO_2_ films (TiO_2_P and TiO_2_G) before and after heating were analysed using XRD. A Bruker D8 Advance apparatus was used to record the XRD patterns with an irradiation source of Co-Kα (λ = 1.789 Å) operating at 35 kV and 28 mA.

#### 2.2.3. Laser Scanning Confocal Microscope (LSCM)

The surface morphology of TiO_2_ films (TiO_2_P and TiO_2_G) was measured using a LSCM (Olympus LEXT OLS5000-SAF 3D LSCM) with 100x/0.80NA and 50x/0.60NA LEXT objective lenses. The Olympus Data Analysis software was used to calculate the arithmetic mean deviation (Ra) and root mean square deviation (Rq) values.

#### 2.2.4. X-ray Photoelectron Spectroscopy (XPS)

XPS analysis was performed using an X-ray source with Mg Kα line (hv = 1253.6 eV). A detailed description of the equipment has been given previously [27]. Survey spectrum scans were performed with a pass energy of 40 eV using a SPECS PHOIBOS-HSA 3500 hemispherical analyser. High-resolution XPS spectra were recorded for C, O, P, Si, Ti, Cr and Au with a pass energy of 10 eV. All XPS binding energy scales were normalised using the C 1 s peak at 285 eV. The peaks were fitted to calculate relative intensities considering atomic sensitivity factors. XPS was recorded immediately after sample preparation and heating, thus, reducing the number of atmospheric exposures.

## 3. Results and Discussion

### 3.1. Influence of the Thickness of the TiO_2_ Films

The influence of the thickness of the RF sputter-deposited TiO_2_ on the change in film morphology upon heating is investigated. First, we will determine the thickness of the TiO_2_ films for TiO_2_P and TiO_2_G and describe the crystallinity and morphology of both samples before and after heating. Then, the XPS results will be reported for both TiO_2_P and TiO_2_G. Subsequently, the agglomeration of Au_9_ clusters beneath a Cr_2_O_3_ overlayer upon heating of the two films is determined and discussed.

### 3.2. Determination of the TiO_2_ Film Thickness

Figure 1 shows cross-section SEM images of TiO_2_P and TiO_2_G with line measurements of the thickness of the TiO_2_ films. These SEM images clearly show that the thickness of the film for the TiO_2_P and TiO_2_G samples is ~400 nm and ~1100 nm, respectively; the film thickness of TiO_2_G is more than two times greater than for TiO_2_P. To confirm the film thickness, EDAX was further processed at the same image spots as SEM. Cross-section SEM-EDAX elemental maps of Ti, O and Si of TiO_2_P and TiO_2_G are shown in Appendix A. The EDAX elemental maps confirm that the thickness of the TiO_2_ film for TiO_2_G is larger than for TiO_2_P.

### 3.3. Crystal Structure of the TiO_2_P and TiO_2_G before and after Heating

To assess the crystal structure of the TiO_2_ film for TiO_2_P and TiO_2_G, XRD was conducted (Figure 2). There are no observable anatase, rutile or brookite crystal phase peaks [29], indicating that the films have an amorphous crystal structure. The crystallographic state of the TiO_2_ is known to be transformed upon heating. The XRD patterns of TiO_2_ films (TiO_2_P and TiO_2_G) after heating at 200 °C for 10 min are shown in Figure 2. Both spectra show an anatase peak at 29.8°, which confirms that the crystal structure of TiO_2_P and TiO_2_G has changed to the anatase phase after heating. The intensity of the anatase diffraction peak for TiO_2_G is more than two times higher than for TiO_2_P, which is due to the difference in the total amount of TiO_2_ in each film. The TiO_2_G layer is more than two times thicker than TiO_2_P, so we also expect that there is more than twice as much anatase in the TiO_2_G film. Thus, the percentage change in crystal structure in the films is comparable. The formation of the anatase phase strongly suggests the TiO_2_ film could be mobile during the heating process, which could influence the morphology of the TiO_2_ films, as will be discussed below.

### 3.4. Morphology of the TiO_2_P and TiO_2_G Layer before and after Heating

LSCM was conducted on both TiO_2_ films before and after heating to compare their morphology. Figure 3 shows the surface morphology of TiO_2_P and TiO_2_G before and after heating over an area of 16 × 16 µm and the determined Ra and Rq values. The 3D profiles of the same spots are displayed in Appendix A. Before heating, the Ra (and Rq) values of the TiO_2_P and TiO_2_G are 0.6 nm (0.8 nm) and 1.0 nm (1.3 nm), respectively. However, after heating, the values become 1.0 nm (1.2 nm) and 12.7 nm (14.7 nm), respectively. The change in Ra (and Rq) for TiO_2_P is small after heating, especially in comparison to TiO_2_G, which is 12 times higher after heating. The Ra (and Rq) values were also calculated over a much larger area of 595 × 595 µm and show a similar change (Appendix A). The change in the Ra (and Rq) values indicates that both the TiO_2_P and TiO_2_G increase in surface roughness after heating. The XRD results show that the TiO_2_G and TiO_2_P have the same fraction of anatase after heating, so the total amount of anatase in TiO_2_G is larger compared to TiO_2_P (vide supra). Çörekçi et al. noted a similar finding in their study of different thicknesses of TiO_2_ films heated at different temperatures [19]. The authors reported that the surface roughness of the thicker TiO_2_ film (300 nm) increased more compared to thinner films (220 and 260 nm) upon heating. In our study, a large change in the surface roughness was observed clearly with the thicker film (more than two times thicker) by a factor of six. Çörekçi et al. assumed that the increase in surface roughness was due to increases in the grain sizes with increasing film thickness and the recrystallization in the TiO_2_ films during heating. A number of studies have reported comparable findings that the surface morphology of the TiO_2_ films changes upon heating [17,30]. Thus, we conclude that the thicker TiO_2_G film is more mobile during heating in comparison to the thinner film in the TiO_2_P sample.

### 3.5. Au_9_ Clusters on TiO_2_P and TiO_2_G; a Probe for Mobility during Heating

In order to provide insight into the mobility of the TiO_2_ during the recrystallisation process, Au_9_ clusters were deposited onto the TiO_2_ films and analysed with XPS. XPS was used to investigate the size of phosphine-protected Au_9_ clusters deposited onto TiO_2_P and TiO_2_G. In addition, the effect of the CrO_x_ overlayer on the Au_9_ clusters was investigated, also with XPS. Figure 4 and Figure 5 show the peak positions and relative intensities of Au 4f_7/2_ peaks in the XP spectra of three different concentrations (0.006, 0.06 and 0.6 mM) of TiO_2_P-Au_9_, TiO_2_G-Au_9_, TiO_2_P-Au_9_-CrO_x_ and TiO_2_G-Au_9_-CrO_x_ before and after heating. Appendix A show a summary of all the Au 4f_7/2_ peak positions and full-width-half-maximum (FWHM). Note that all the Au 4f spectra for both substrates (TiO_2_P and TiO_2_G) are shown in Appendix A. The TiO_2_P XPS results will be first presented, followed by the TiO_2_G results.

### 3.6. XPS of TiO_2_P Sample

Without the CrO_x_ layer and before heating, the Au 4f_7/2_ peaks appeared at 85.1–85.4 eV with an FWHM of 1.7–1.8 eV (Figure 4A), whereas after heating, the Au 4f_7/2_ peaks shifted to slightly lower binding energies (84.7–84.8 eV) and FWHM (1.5–1.6 eV), and also showed a decrease in relative Au intensity across all Au_9_ concentrations (Figure 4B). The results of the samples covered with a CrO_x_ layer are shown in Figure 4C,D. The Au 4f_7/2_ peak positions of TiO_2_P-Au_9_ after CrO_x_ deposition but before heating were observed at 85.3 eV and an FWHM of 1.6 eV for all three concentrations. Note that the Au relative intensities decrease after the photodeposition of the CrO_x_ layer, confirming the coverage of Au clusters with the CrO_x_ layer (Figure 4D). After heating, the XPS peak position decreases slightly to 85.0 eV with no significant change in FWHM. The relative Au intensities also remained unchanged upon heating. XPS has been shown previously to be a reliable indicator of the size of phosphine-protected Au_9_ clusters through the final state effect [21,28,31,32,33,34,35,36]. Generally, non-agglomerated Au_9_ clusters on TiO_2_ appear at a high binding peak (HBP) between 85.0–85.4 eV with an FWHM of 1.7 ± 0.2 eV, and agglomerated Au_9_ clusters shift toward a low binding peak (LBP) at 84 eV with a decreasing FWHM that corresponds to bulk Au [28,31,32,33,34,35]. This XPS interpretation has been confirmed by correlating the XPS results with other techniques, such as HRTEM [33,34]. Here, the Au 4f_7/2_ peak positions of TiO_2_P-Au_9_ without the CrO_x_ layer after heating indicate a small degree of agglomeration of the Au_9_ clusters for all concentrations. This is further confirmed by a small decrease in Au intensity after heating, indicating that some of the gold is attenuated due to some larger, agglomerated particles. Electrons emitted from the part of the clusters facing toward the substrate are attenuated when leaving the sample, which decreases the overall Au intensity [31,32]. Therefore, the same total amount of gold deposited on the surface will have a lower intensity for large gold particles than that of small gold clusters. In contrast to the CrO_x_ layer of the Au 4f_7/2_ peaks, positions are unchanged after heating and there is no further decrease in the Au relative intensities, indicating that Au clusters remain non-agglomerated clusters with CrO_x_ coverage (see Figure 1A). It is important to note that there is a decrease in Au intensity after photodeposition of the CrO_x_ layer due to the coverage of Au_9_ clusters (Figure 4D). These results are in agreement with our previous report showing that CrO_x_ overlayers inhibit the agglomeration of Au clusters [28].

The P 2p spectra of TiO_2_P-Au_9_ without and with the CrO_x_ layer before and after heating are shown in Appendix A and the peak positions are discussed in the Appendix A. The Cr 2p spectra for TiO_2_P-Au_9_-CrO_x_ before and after heating at the three different concentrations are shown in Appendix A. A summary of all the Cr 2p_3/2_ peak positions is shown in Appendix A and the peak positions are discussed in the Appendix A.

### 3.7. XPS of TiO_2_G Sample

For the thicker film, TiO_2_G-Au_9_, the Au 4f_7/2_ peak positions before heating for all three different concentrations appeared at the HBP at 85.3 ± 0.1 eV (Figure 5A) and an FWHM of 1.8 ± 0.2 eV, corresponding to non-agglomerated Au clusters. However, after heating, the Au 4f_7/2_ shifted toward lower energy (84.6–84.9 eV) and an FWHM of 1.5–1.7 eV with a decrease in Au intensity (Figure 5B), indicating that Au clusters are partially agglomerated. With the CrO_x_ layer deposited before heating, the Au 4f_7/2_ peak positions are observed at the HBP position at 85.3–85.5 eV (Figure 5C), with a decrease in Au 4f_7/2_ intensity due to the coverage of the CrO_x_ layer on Au_9_ clusters (Figure 5D). There is a slight increase in the binding energy of the Au 4f peak after the photodeposition of CrO_x_, and we do not know if this is a significant change or not. However, the position found can be used as an indication of the presence of non-agglomerated Au clusters. With the CrO_x_ layer after heating, the Au 4f_7/2_ peak positions have further shifted to lower energy (84.3–84.8 eV) positions and an FWHM of 1.3–1.8 eV with a decrease in Au intensity, which is attributed to further agglomeration of the Au clusters based on the final state effect (see Figure 1B). The degree of agglomeration increases with increasing Au_9_ concentration for both cases (without and with the CrO_x_ layer). Note here the difference; Au clusters on the surface of TiO_2_G undergo increased agglomeration after heating, even in the presence of the CrO_x_ layer. This is different to the TiO_2_P, where Au clusters are less likely to agglomerate under the CrO_x_ layer after heating. This difference will be further discussed below.

The chemical state of the phosphorous ligands of TiO_2_G-Au_9_ without and with the CrO_x_ layer, both before and after heating, was determined using the P 2p region (see Appendix A for more information and accompanying text). Appendix A shows the Cr 2p spectra for TiO_2_G-Au_9_-CrO_x_ before and after heating of the three different concentrations. All the Cr 2p_3/2_ peak positions are given in Appendix A and the peak positions are discussed in the Appendix A.

### 3.8. Effect of the TiO_2_ Film Thickness

The protective effect of the CrO_x_ layer on the agglomeration of Au_9_ clusters is not the same for both the TiO_2_P and TiO_2_G substrates. The agglomeration of Au_9_ clusters is inhibited on TiO_2_P with the CrO_x_ overlayer but not on TiO_2_G, which shows a higher degree of agglomeration. The coverage of the CrO_x_ layer on Au_9_ clusters for both substrates is demonstrated by the decrease in the Au-XPS intensities. After heating, it is observed that the relative amount of CrO_x_ decreases for both films (Appendix A). Our previous studies on a similar system revealed that the CrO_x_ layer diffuses into a TiO_2_ film after heating to 600 ℃ due to the differences in surface energy between TiO_2_ and CrO_x_ [26]. In this study, both films were heated to only 200 ℃, however, CrO_x_ on TiO_2_G experienced more diffusion of CrO_x_ into the film compared to TiO_2_P. One possibility for the higher degree of Au_9_ agglomeration and CrO_x_ diffusion is the mobility of the TiO_2_ film. Cluster agglomeration can be due to either (i) growth of the clusters over the surface or (ii) mobility of the substrate. In the case of (i), the cluster growth and agglomeration on a substrate can be ascribed to either Smoluchowski ripening or Ostwald ripening mechanisms. For Smoluchowski ripening, the agglomeration of clusters is caused by the collision and coalescence of entire clusters to larger particles [37]. For Ostwald ripening, the growth of larger particles takes place by the detachment of single atoms, which diffuse onto a nearby cluster or nanoparticle [38]. In the case of (ii), a section of the substrate to which a cluster is adsorbed moves closer to another section of the substrate, which has another adsorbed cluster. The significant change in the surface morphology of TiO_2_G after heating (Ra: 11.7 nm and Rq: 13.4 nm) compared to TiO_2_P (Ra: 0.4 nm and Rq: 0.5 nm) strongly suggests that the agglomeration of the Au_9_ clusters with different concentrations on TiO_2_G after heating is due to the high distortion of the surface upon heating. A higher mobility of the TiO_2_ substrate during heating means that the local surface beneath an Au cluster moves larger distances compared to a substrate which exhibits lower mobility during heating (see Figure 2). The high mobility of the thick film is assumed to be due to the recrystallisation during heating, which is in agreement with previous studies [17,19,30]. With increasing mobility, the likelihood of close contact between two or more Au clusters increases, and thus the likelihood of agglomeration is also increased. Furthermore, the degree of agglomeration of the Au clusters is larger for the thicker TiO_2_G substrate compared to the thinner TiO_2_P substrate.

## 4. Conclusions

In summary, the change in surface morphology of two different film thicknesses of RF sputter-deposited TiO_2_ (~400 nm and ~1100 nm) was examined and compared upon heating. After heating, the thick TiO_2_ film showed a larger change in surface morphology, which is associated with higher mobility during heating compared to the thin TiO_2_ film. The difference in mobility is attributed to the differences in the total amount of amorphous TiO_2_ transformed to anatase in each of the films, which then results in differences in the morphology of the surface upon heating. Au_9_ clusters were used as a probe for TiO_2_ mobility. Au_9_ clusters were deposited onto the two different TiO_2_ films, followed by photodeposition of the CrO_x_ layer. After heating, the Au clusters on the thicker film showed a larger degree of agglomeration compared to the thinner film. The higher mobility of the thick film during heating increased the probability of close encounters of Au clusters, which resulted in agglomeration of the Au_9_ clusters even in the presence of a CrO_x_ overlayer. In contrast, the lower mobility of the thin film resulted in less agglomeration of the Au_9_ clusters after heating.

## Data Availability

The data that support the findings of this study are available from the corresponding author upon reasonable request.

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
