# Peer review of "Effect of TiO2 Film Thickness on the Stability of Au9 Clusters with a CrOx Layer"

_nanomaterials, 2022, doi:10.3390/nano12183218_

Round 1
Reviewer 1 Report
The authors have used titania film to be coating with Au nanoclusters and then they have stabilized with CrOx. My question is, how have they made sure that the Au nanoclusters are synthesized without having any proof even a reference in the XPS, showing that the Au there is cluster. In the XPS, after coating with CrOx, a change in the peak position of Au has also occurred while the authors have not mentioned this, but to the intensity of the peaks. There is not enough characterization of the catalyst, making the paper unable to publish in the journal. Authors should provide more details and characterization of materials and the procedure.
My minor comments, are also below:
The figures and plots are of low quality and should be improved.
The introduction should be improved by pointing to the recent advances in the field of TiO2:
Applied Catalysis B: Environmental, 2021, 297, 120380; Journal of Membrane Science, 489, 43-54. DOI: 10.1016/j.memsci.2015.04.010.
Author Response
Response to Reviewers’ Comments: nanomaterials-1895874
The authors would first like to thank the reviewers for taking their time in reading the article and offering feedback. We have made the required changes and addressed the comments in the article and have listed them in this document.
In our response, we first repeat the comment from the reviewer and editors and then add our response in blue. The changes are marked in red in the manuscript and SI files. We first address the editor's comments and the reviewers’ comments.
Reviewer 1
The authors have used titania film to be coating with Au nanoclusters and then they have stabilized with CrOx. My question is, how have they made sure that the Au nanoclusters are synthesized without having any proof even a reference in the XPS, showing that the Au there is cluster.
Reply: In the last paragraph on page 7 we describe the XPS data and refer to a range of publications which all provide evidence how to identify from XPS the size of the Au clusters. To be more comprehensive, we have added a UV-Vis spectrum of the synthesized Au9(PPh3)8(NO3)3 nanoclusters to the supplementary information (Figure S2) and refer to this figure in paragraph 4 on page 3.
In the XPS, after coating with CrOx, a change in the peak position of Au has also occurred while the authors have not mentioned this, but to the intensity of the peaks.
Reply: We had already discussed on page 9 that “There is a slight increase in the binding energy of the Au 4f peak after the photodeposition of CrOx, and we do not know if this a significant change or not. However, the peak position can be used as an indication of the presence of non-agglomerated Au clusters”
There is not enough characterization of the catalyst, making the paper unable to publish in the journal. Authors should provide more details and characterization of materials and the procedure.
Reply: This comment might be based on a misunderstanding. In the last paragraph on page 2 we describe that the Au clusters are used here as a probe for monitoring the change in surface morphology upon hearing. This publication does not relate to catalysis as such although the Au clusters could be used as a co-catalyst.
My minor comments, are also below:
1- The figures and plots are of low quality and should be improved.
Reply: We appreciate the suggestion of the reviewer. The quality and clarity of the figures have been improved throughout the manuscript.
- The introduction should be improved by pointing to the recent advances in the field of TiO2:
Applied Catalysis B: Environmental, 2021, 297, 120380; Journal of Membrane Science, 489, 43-54. DOI: 10.1016/j.memsci.2015.04.010.
Reply: We have reviewed the suggested papers and following the editor's guidance “(I) Please check that all references are relevant to the contents of the manuscript.”, contend that these papers are not adequately relevant to the manuscript and therefore have not been included.
Further changes: the grammar of the manuscript has been changed at a few places to improve the readability.
Reviewer 2 Report
In this manuscript, the authors reported the effect of TiO2 film thickness on the stability of Au9 clusters with a CrOx layer by using RF magnetron sputtering method. The work should be interested and helpful to other researchers. The manuscript can be considered publishing in nanomaterials but only after addressing the following issues properly.
1、How to control the thickness of the film, and why choose these two thicknesses for research?
2、Check the Figure sequence number in Figure 1. The font in the figures shall be uniform.
3、“The intensity of the anatase diffraction peak for TiO2G is more than two times higher than for TiO2P, which is due to the difference in the total amount of TiO2 in each film.(P5)” What is the reason?
4、Authors used RF magnetron sputtering method to prepare samples, but authors did not explain its importance. What are the highlights and features of this article? Authors should strengthen the logic of the introduction.
5、RF is also widely used in plasma field, like Chemical Engineering Journal 450 (2022) 138225, authors can cite.
Author Response
Response to Reviewers’ Comments: nanomaterials-1895874
The authors would first like to thank the reviewers for taking their time in reading the article and offering feedback. We have made the required changes and addressed the comments in the article and have listed them in this document.
In our response, we first repeat the comment from the reviewer and editors and then add our response in blue. The changes are marked in red in the manuscript and SI files. We first address the editor's comments and the reviewers’ comments.
Reviewer 2
In this manuscript, the authors reported the effect of TiO2 film thickness on the stability of Au9 clusters with a CrOx layer by using RF magnetron sputtering method. The work should be interested and helpful to other researchers. The manuscript can be considered publishing in nanomaterials but only after addressing the following issues properly.
- How to control the thickness of the film, and why choose these two thicknesses for research?
Reply: The thickness of RF sputter deposited TiO2 can be controlled by various parameters such as the sputtering power and time. We have now added in the second paragraph of the introduction a sentence how to control the thickness of TiO2. As outlined in the second paragraph on page 3, the two thicknesses used were a result of the chosen conditions and not specifically targeted.
- Check the Figure sequence number in Figure 1. The font in the figures shall be uniform.
Reply: We follow with the reviewer’s suggestion and have revised all the figures with uniform font.
- “The intensity of the anatase diffraction peak for TiO2G is more than two times higher than for TiO2P, which is due to the difference in the total amount of TiO2 in each film.(P5)” What is the reason?
Reply: On page 5 it is stated that the thickness of TiO2G is two times greater than for TiO2P so the intensity of the anatase peak in the XRD should be twice assuming that the same proportion of amorphous TiO2 is transformed for each film upon heating under the same conditions.
- Authors used RF magnetron sputtering method to prepare samples, but authors did not explain its importance. What are the highlights and features of this article? Authors should strengthen the logic of the introduction.
Reply: We appreciate the comment of the reviewer. We have revised the introduction and added more references. Also, we have added in the fourth paragraph of the introduction a sentence highlighting the importance of this article.
- RF is also widely used in plasma field, like Chemical Engineering Journal 450 (2022) 138225, authors can cite.
Reply: We appreciate the suggestion of the reviewer. However, after reviewing the suggested paper and by following the editor's guidance “(I) Please check that all references are relevant to the contents of the manuscript.”, we contend that this paper does not add to the focus of this manuscript.
Further changes: the grammar of the manuscript has been changed at a few places to improve the readability.
